# Functional Characterisation of the Poplar Atypical Aspartic Protease Gene *PtAP66* in Wood Secondary Cell Wall Deposition

Shenquan Cao, Cong Wang, Huanhuan Ji, Mengjie Guo, Jiyao Cheng, Yuxiang Cheng * and Chuanping Yang *

State Key Laboratory of Tree Genetics and Breeding, Northeast Forestry University, Harbin 150040, China; shenquancao@gmail.com (S.C.); cc1997121@163.com (C.W.); HuanJ0331@163.com (H.J.); mengjieguo0327@gmail.com (M.G.); chengjiyao1993@163.com (J.C.)

* Correspondence: chengyuxiang@nefu.edu.cn (Y.C.); yangcp@nefu.edu.cn (C.Y.);
Tel.: +86-451-82192153 (Y.C.); +86-451-82191086 (C.Y.)

**Abstract:** Secondary cell wall (SCW) deposition is an important process during wood formation. Although aspartic proteases (APs) have been reported to have regulatory roles in herbaceous plants, the involvement of atypical APs in SCW deposition in trees has not been reported. In this study, we characterised the *Populus trichocarpa* atypical AP gene *PtAP66*, which is involved in wood SCW deposition. Transcriptome data from the AspWood resource showed that in the secondary xylem of *P. trichocarpa*, *PtAP66* transcripts increased from the vascular cambium to the xylem cell expansion region and maintained high levels in the SCW formation region. Fluorescent signals from transgenic *Arabidopsis* plant roots and transiently transformed *P. trichocarpa* leaf protoplasts strongly suggested that the PtAP66-fused fluorescent protein (PtAP66-GFP or PtAP66-YFP) localised in the plasma membrane. Compared with the wild-type plants, the Cas9/gRNA-induced *PtAP66* mutants exhibited reduced SCW thickness of secondary xylem fibres, as suggested by the scanning electron microscopy (SEM) data. In addition, wood composition assays revealed that the cellulose content in the mutants decreased by 4.90–5.57%. Transcription analysis further showed that a loss of *PtAP66* downregulated the expression of several SCW synthesis-related genes, including cellulose and hemicellulose synthesis enzyme-encoding genes. Altogether, these findings indicate that atypical *PtAP66* plays an important role in SCW deposition during wood formation.

**Keywords:** atypical aspartic protease; Cas9/gRNA; *Populus trichocarpa*; *PtAP66*; secondary cell wall; subcellular localisation; wood formation

## 1. Introduction

Wood is a product with abundant biomass for carbon neutrality in terrestrial ecosystems, and it is also an important sustainable and renewable resource for biofuels and raw materials [1]. Its formation is a complex biological process, including cell division of the vascular cambium, xylem cell expansion, secondary cell wall (SCW) deposition, and programmed cell death [2]. The construction of wood mainly occurs by thickened SCWs, which begin to deposit after xylem cell expansion ceases and largely consist of cellulose, hemicellulose, and lignin. The exploitation degree of wood resources is largely determined by the structure and chemical components of the SCW produced by plants [3]. Therefore, research on the regulation of SCW deposition in woody plants is vital for biomass accumulation and utilisation.

SCW deposition requires the coordinated expression of various carbohydrate active enzymes [4–6], and this process is controlled by a multiple layer transcription factor regulation network [7–10]. In fact, SCW deposition is also dynamically regulated by various genetic and environmental signals [2]. Concerning hormone signals, auxin, gibberellin, and brassinosteroid, which affect the expression of SCW-related genes, have been implicated in regulating SCW deposition in woody plants [11–14]. Moreover, oligosaccharides and reactive oxygen species (ROS) have been reported to be involved in the regulation of

SCW deposition by modulating the transcriptional regulatory programme and ROS homeostasis [15,16], respectively. In addition, a variety of regulators, including cysteine protease, receptor-like kinase, and osmotin-like protein, have also been identified in connection with SCW deposition [17–19]. Environmental signals, such as gravity, drought, high salinity, and light irradiance, also regulate SCW deposition [20–23]. Although emerging evidence has revealed various signals implicated in the regulation of SCW deposition, comprehensive pathways have not been fully established and whether other signalling pathways exist remains to be explored.

Aspartic proteases (APs) are one of the major proteolytic enzyme families in plants, and they are involved in many aspects of growth and development. In general, plant APs can be divided into three groups: typical, atypical, and nucellin-like APs [24]. Among them, atypical APs have been implied to have diverse signal regulatory roles in stress responses, chloroplast metabolism, and reproductive processes [25]. For example, CDR1 (*constitutive disease resistance*) encodes an extracellular atypical AP, and plants that overexpress this *AP* exhibit constitutive expression of defence-related genes [26]. AED1 (APOPLASTIC, ENHANCED DISEASE SUSCEPTIBILITY1-DEPENDENT1) encodes another apoplastic atypical AP, and simultaneous suppression of *AED1* and its neighbouring locus (*At5g10770*, an atypical AP) causes a stunted phenotype [27]. NANA encodes a chloroplast atypical AP, and misexpression of *NANA* influences photosynthetic carbon metabolism and nuclear gene expression [28]. In addition, the endoplasmic reticulum protein atypical ASPG1 (ASPARTIC PROTEASE IN GUARD CELL 1) was proposed to degrade seed storage proteins that affect seed quality and seed germination [29]. To date, only nine plant atypical APs have been investigated and partially characterised, and these studies were mainly limited to herbaceous plants. Thus, the functions of the vast majority of the members in atypical AP groups are still unknown. Considering that some atypical APs have been found in the stems of woody plants [30,31], they might play a key regulatory function in SCW deposition.

Due to its fast growth rate and large wood size, poplar is an excellent tree species for afforestation and biomass accumulation [32–34]. A complete genome sequence has been identified for *Populus trichocarpa* [35], which facilitates the exploration of gene functions associated with SCW deposition. In our previous study, a number of atypical *AP* genes were shown to be expressed in *P. trichocarpa* xylem, and particularly, the promoter activity of *PtAP66* was highly evident in xylem fibres [30]. This specific expression pattern suggested that the function of *PtAP66* might be associated with wood formation. In this study, we used CRISPR/Cas9 technology to produce *ptap66* mutant trees. Genetic and phenotypic analyses indicated that *PtAP66* is involved in SCW deposition. This work provides novel insights into the role of atypical APs in tree growth and development.

## 2. Materials and Methods

### 2.1. Plant Materials and Growth Conditions

The *P. trichocarpa* (Nisqually-1) used in this study was obtained from the greenhouse of the Northeast Forestry University, China. The conditions for plant cultivation were described in a previous study [36]. The plantlets were transplanted into soil for the phenotypic analysis in a growth greenhouse (22–25 °C, 18 h light/6 h dark cycle, and ~250 $\mu$mol m$^{-2}$ s$^{-1}$ light intensity). *Arabidopsis thaliana* (ecotype Columbia) was grown on 1/2 Murashige and Skoog (MS) medium (Phytotech, Lenexa, KS, USA) plates containing 1% (*w/v*) sucrose at pH 5.7 under long-day conditions (22 °C, 16 h light/8 h dark, and 80–120 $\mu$mol m$^{-2}$ s$^{-1}$ light intensity) in a chamber. Seedlings were transferred from the MS plates to soil for transformation in a greenhouse under the same conditions.

### 2.2. Vector Construction

*PtAP66*-TOPO was obtained from our previous study [30], and this entry clone was then constructed into the pGWB5 and p2GWY7 vectors by the Gateway LR reaction (Invitrogen, Carlsbad, CA, USA) to generate *35S::PtAP66-GFP* (green fluorescent protein)

and *35S::PtAP66-YFP* (yellow fluorescent protein) constructs, respectively. The *35S::GFP* and *35S::YFP* constructs were obtained from previous studies [37,38]. For the Cas9/gRNA (pHSE401-2gRNA) constructs, each target site was selected by manual inspection and searched against the *P. trichocarpa* genome to ensure target specificity. The potential targets were as close as possible to the 5′-ends of the coding sequence (CDS), with the objective that the induced mutations would produce translation terminators early. The entire construction process was performed according to our previous protocol [39]. All destination vectors were sequenced and then introduced into *Agrobacterium tumefaciens* strain GV3101 for transformation. The primers are listed in Table S1.

### 2.3. Generation of Transgenic Plants and Detection of Cas9/gRNA-Induced Mutations

*A. thaliana* transformation was performed by the floral-dip method [40]. The positive transformants were selected on MS plates containing 50 µg mL$^{-1}$ kanamycin, and several transgenic lines with high transcription levels of *GFP* or *PtAP66* were further propagated. For the *P. trichocarpa* transformation, the *A. tumefaciens* strains that contained the pHSE401-2gRNA vectors were incubated to an OD600 value of 0.4–0.6 for bacteria collection, and then these bacterial pellets were resuspended to an OD600 value of 0.4 for plantlet stem infection. The detailed operation processes, including screening and rooting of transformants, were performed according to a previous protocol [36].

*P. trichocarpa* genomic DNA was extracted using a plant genomic DNA extraction kit (Bioteke, Beijing, China). The genomic DNA fragments flanking the four gRNA target sites of *PtAP66* were cloned by polymerase chain reaction (PCR) amplification. The PCR assay was performed using Ex*Taq* (TaKaRa, Dalian, China) in a total volume of 50 µL solution, and the parameters were 94 °C for 3 min; followed by 35 cycles of 94 °C for 30 s, 60 °C for 60 s, and 72 °C for 60 s; and a final extension of 72 °C for 7 min. The DNA fragments were purified by a Silica Bead DNA Gel Extraction Kit (Thermo, Shanghai, China) and cloned into pMD18-T vector (TaKaRa, Dalian, China), and at least 20 positive clones for each line were selected for sequencing. The primers are listed in Table S1.

### 2.4. Subcellular Localisation

To analyse the subcellular localisation of PtAP66, the five-day-old transgenic *Arabidopsis* seeding roots overexpressing *PtAP66-GFP* or *GFP* were used to observe GFP fluorescence under a confocal laser scanning microscope (LSM 700, Zeiss, Jena, Germany) with a 488 nm excitation laser. Some plant roots were induced to plasmolysis by treatment with 1 M mannitol for 5 min at room temperature as previously described [41]. Additionally, we used leaf protoplasts to observe the subcellular localisation of PtAP66-YFP. The isolation and transformation of *Populus* leaf protoplasts were performed according to previous protocols [42,43] with some modifications. In brief, 1 mm fine leaf strips from 4-week-old sterile plantlets were digested in enzyme solution (3% (*w/v*) cellulase R10 (Yakult Pharmaceutical Ind. Co., Ltd., Tokyo, Japan), 0.8% (*w/v*) macerozyme R10 (Yakult Pharmaceutical Ind. Co., Ltd., Tokyo, Japan), 400 mM mannitol, 20 mM KCl, 20 mM MES (pH 5.7), 10 mM CaCl$_2$, and 0.1% (*w/v*) bovine serum albumin (BSA)). After vacuum infiltration of the leaf strips for 30 min, these immersed strips were digested continuously for 4 h without shaking in the dark at room temperature. The enzyme/protoplast solution was diluted with an equal volume of W5 buffer (2 mM MES (pH 5.7), 154 mM NaCl, 125 mM CaCl$_2$, and 5 mM KCl), and the liberative protoplasts were harvested by filtering through a 70 µm pore nylon cloth. Then, the collected protoplasts were resuspended in MMG buffer (400 mM mannitol, 15 mM MgCl$_2$, and 4 mM MES (pH 5.7)) to a concentration of $1 \times 10^5$ protoplasts mL$^{-1}$. Next, 110 µL resuspended protoplasts was transfected with 10–20 µg of each plasmid DNA (1 µg µL$^{-1}$), and then 120 µL transfection solution (40% (*w/v*) polyethylene glycol 4000, 200 mM mannitol, and 100 mM CaCl$_2$) was added. The mixtures were incubated at room temperature for 5–10 min before 440 µL of W5 solution was added to terminate the reaction. The leaf protoplasts were gently resuspended with 1 mL WI solution (500 mM mannitol, 20 mM KCl, and 4 mM MES (pH 5.7)). After incubation at room temperature for 12–16 h,

yellow fluorescent signals were visualised with a confocal laser scanning microscope as described above.

### 2.5. Growth and Xylem Cell Morphology Analysis

Wild-type (WT) and *ptap66* mutant plantlets were transplanted into turfy soil and grown in the greenhouse. After 30 days, normal plants (six individual plants for each line) were marked at 20 cm above the soil, and these points were used as the reference for analysing plant growth. The tree heights were measured from the apical buds to the reference points every month over a period of 4 months. The stem diameters and internode numbers were counted, with the first stem internode (IN1) defined as that below the first fully expanded leaf in the apex. The average of stem internodes (IN10, IN11, and IN12) with the leaves was used to measure the length of the internodes as well as the length and width of the leaves. For the width analysis of transverse stem tissues, the IN12 of poplars were prepared and four random positions per stem section and 12 measurements per tree were recorded by an upright microscope (BX43, Olympus, Shinjuku, Japan). For the fibre size measurement, the peeled IN22 was macerated in a solution containing 10% nitric acid and 10% chromic acid at 60 °C for 4–6 h and then rinsed with distilled water. After rinsing, the stem segments were soaked again in the proper amount of sterile water, and the outer cells were separated by shaking gently. The suspension fibres were stained with equal volumes of 0.1 % acid fuchsin and placed on the slides, then photographed by an upright microscope (BX43, Olympus, Shinjuku, Japan). The physical length and width of at least 300 fibres per tree were measured.

### 2.6. Scanning Electron Microscopy (SEM) and Wood Composition Assay

Stem segments were prepared for the SEM analysis as previously described [44]. Briefly, the IN12 of the WT and *ptap66* plants were fixed conventionally in formalin-acetic acid-alcohol solution (FAA; 50% ethanol, 3.65% formaldehyde, and 5% glacial acetic acid) overnight and dehydrated in a graded series of ethanol (50%, 60%, and 70%, 2 h each step). These free-hand cross-sections of stem segments were dried naturally, coated with gold at 10 mA for 60 s, and finally transferred to a benchtop SEM chamber (JCM-5000, JEOL, Tokyo, Japan) to analyse the SCW thickness of mature xylem fibres at 15 kV. Four sections were analysed per tree, and 40 measurements were recorded by ImageJ. Three individual plants of each line were used for analysis. For the wood composition assay, which was performed as described in our previous protocol [39], the basal stems from 4-month-old trees were peeled and dried at 55 °C, and then the dried tissues were ground to a fine powder. Subsequently, after weighing 70 mg powder per tree for washing (70 % aqueous ethanol, chloroform/methanol (1:1 *v/v*) solution and acetone successively), the obtained insoluble residues were air-dried and prepared as the cell wall materials. We weighed 1.5 mg of cell wall materials per sample for the crystalline cellulose assay, and 2 mg of cell wall materials per sample for lignin content assays. Three samples per tree and three trees from each line were prepared for each composition assay.

### 2.7. RNA Extraction and RT-qPCR Analyses

Total RNA was extracted from the xylem tissues of 3-month-old *P. trichocarpa* plants by plant RNA extraction reagent (Bio-Flux, Beijing, China). cDNA was synthesised using the PrimeScript RT reagent Kit (TaKaRa, Dalian, China) according to the manufacturer's instructions. All quantitative real-time PCR (RT-qPCR) analyses were carried out in a TOWER³G Real-Time PCR System (Jena, Germany) using TB Green *Premix Ex Taq* (TaKaRa, Dalian, China). The PCR programme was performed as described [30]. *PtActin2* was used as the reference, and the $2^{-\Delta\Delta Ct}$ method [45] was used to calculate gene expression levels. The primers are listed in Table S1.

*2.8. Statistical Analysis*

All statistical tests and data analyses were performed using SPSS 19.0 (Chicago, IL, USA). Values are the means ± SD, and asterisks indicate statistical significance at different levels (* $p < 0.05$ and ** $p < 0.01$).

## 3. Results

*3.1. PtAP66 Is Highly Expressed across Wood-Forming Tissues in Poplar*

Our previous report identified 67 *PtAPs* in the *P. trichocarpa* genome, among which 7 atypical *PtAPs* are highly expressed in stem tissues [30], suggesting that these members might play a diverse role in wood formation. Based on their expression patterns [30], we first focused on the *PtAP66* gene, which is highly and preferentially expressed in the developing xylem. We examined the expression of *PtAP66* at different stages of wood formation using the AspWood resource [46]. The data showed that the *PtAP66* transcripts increased dramatically from the vascular cambium to the xylem cell expansion zone and maintained high levels in the SCW formation zone (Figure 1), indicating that *PtAP66* should play a role in wood formation. In contrast, its segmental duplication gene *PtAP5* showed peak expression in the xylem cell expansion zone and a low level of expression over the mature xylem (Figure 1). This result was consistent with previous RT-qPCR expression data and promoter activity analyses [30].

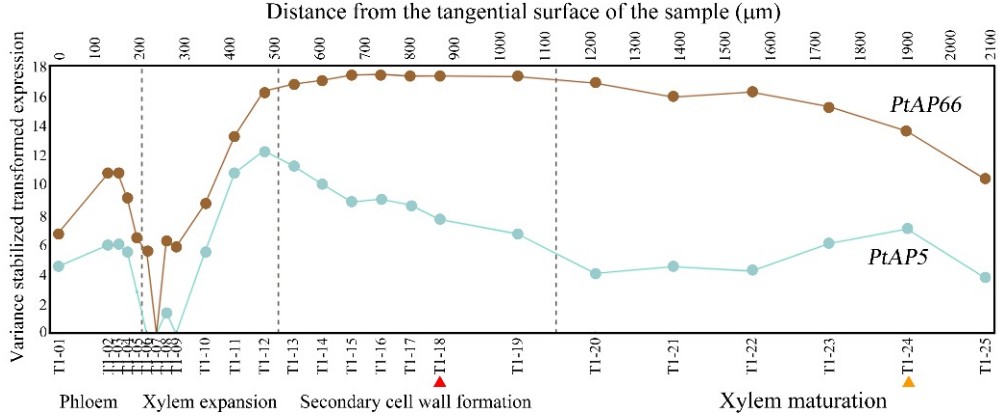

**Figure 1.** Expression profiles of *PtAP66* and *PtAP5* in *Populus* stem tissues.

Gene expression is shown across the stem secondary tissue derived from the AspWood database (http://aspwood.popgenie.org, accessed on 31 March 2021). Dots were generated using the values for the cryosection of stem tissues. The red and orange triangles mark the approximate locations of xylem vessel and fibre cell death as described in a previous study [46].

*3.2. Subcellular Localisation of PtAP66*

PtAP66 belongs to the atypical AP group, which has 54 members in *P. trichocarpa* [30]. PtAP66 was deduced to be a 439-amino-acid protein with an ASP domain (PF00026) located at amino acids 96 to 434. Compared with other plant APs, PtAP66 does not have a plant-specific insert (PSI) domain (Figure S1), which is a vacuolar sorting signal [47]. Additionally, PtAP66 contained a signal peptide at the N-terminus based on the prediction analysis [30] but did not have any canonical subcellular localisation signalling sequences, such as the RGD, C-terminus K/HDEL, or FAEA motifs [47–49].

To determine the PtAP66 subcellular location, we constructed and expressed the *35S::PtAP66-GFP* vector in *Arabidopsis*. Transgenic plant roots were used to observe the PtAP66-GFP location. The fluorescent signals of PtAP66-GFP were mainly localised in the plasma membrane (PM) of root epidermal cells, while the GFP control were dispersed in the cytoplasm and nuclear region (Figure 2a). To test whether PtAP66-GFP is a cell wall

or PM protein, these plant roots were incubated with 1 M mannitol to induce plasmolysis. The PtAP66-GFP fluorescent signals were still dispersed in the PM of the shrink protoplasts (Figure 2a), indicating that the PtAP66-GFP protein localises in the PM. To confirm this localisation pattern, we constructed a fusion vector of *PtAP66* with *YFP* and transiently transformed this chimaeric gene into *P. trichocarpa* leaf protoplasts. The fluorescent signals were similarly observable in the PM, while the control YFP signals appeared ubiquitously in the cytoplasm of protoplasts (Figure 2b). Overall, these findings strongly suggest that PtAP66 localises in the PM.

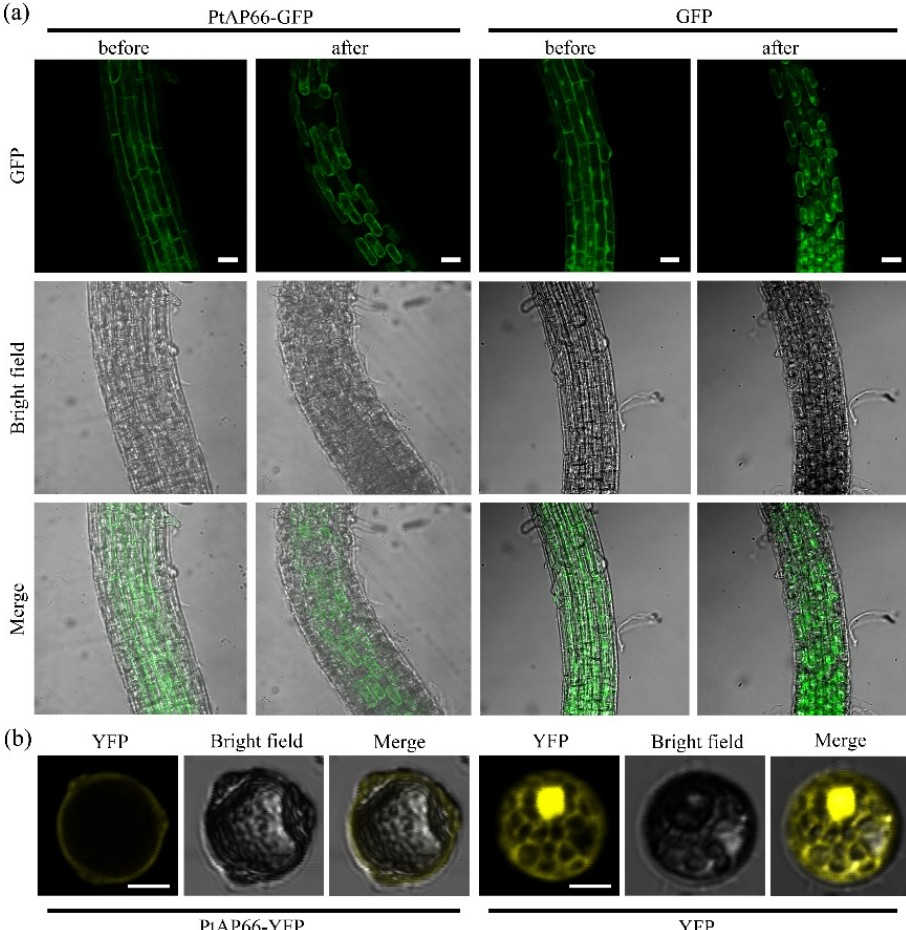

**Figure 2.** Subcellular localisation of the PtAP66-fused fluorescent protein in *Arabidopsis* root cells and *Populus* leaf protoplasts. (**a**) Stable expression of the PtAP66-GFP fusion protein in *Arabidopsis* root cells; and expression of GFP protein as a control. After 1 M mannitol treatments for 5 min, the GFP fluorescence signals were recorded again. Top panels, GFP fluorescence images; middle panels, bright-field images; and bottom panels, merged bright-field and GFP fluorescence images. Bar = 30 μm. (**b**) Subcellular localisation of the PtAP66-YFP fusion protein after transient transformation in *Populus* leaf protoplasts. The expression of YFP protein was used as a control. Bar = 10 μm.

### 3.3. Production of Cas9/gRNA-Induced Ptap66 Mutants

To investigate the function of *PtAP66* in *Populus*, we generated Cas9/gRNA-induced mutations in *PtAP66*. Two pairs of guide RNAs (gRNAs) were selected near the 5′-ends of the CDS (Figure 3a). After genetic transformation, a total of 12 independent transgenic lines were generated, and the DNA fragments flanking the gRNA target sites were amplified from these transgenic lines and WT plants for sequencing analysis. The results revealed that nucleotide deletions (−) and insertions (+) were diverse at the four gRNA target sites (Figure 3b; Table S2) and resulted in premature translation termination of PtAP66 (Figure 3c). As a result, three lines (*ptap66-3*, *ptap66-4* and *ptap66-5*) with biallelic editing in

*PtAP66* were selected and clonally propagated for further phenotypic analysis. PtAP66 and PtAP5 present high amino acid identity (88.8%); thus, we also detected whether the mutations occurred in similar target sites of *PtAP5* in these transgenic lines. The sequencing data showed no mutations in *PtAP5* in these transgenic lines (data not shown).

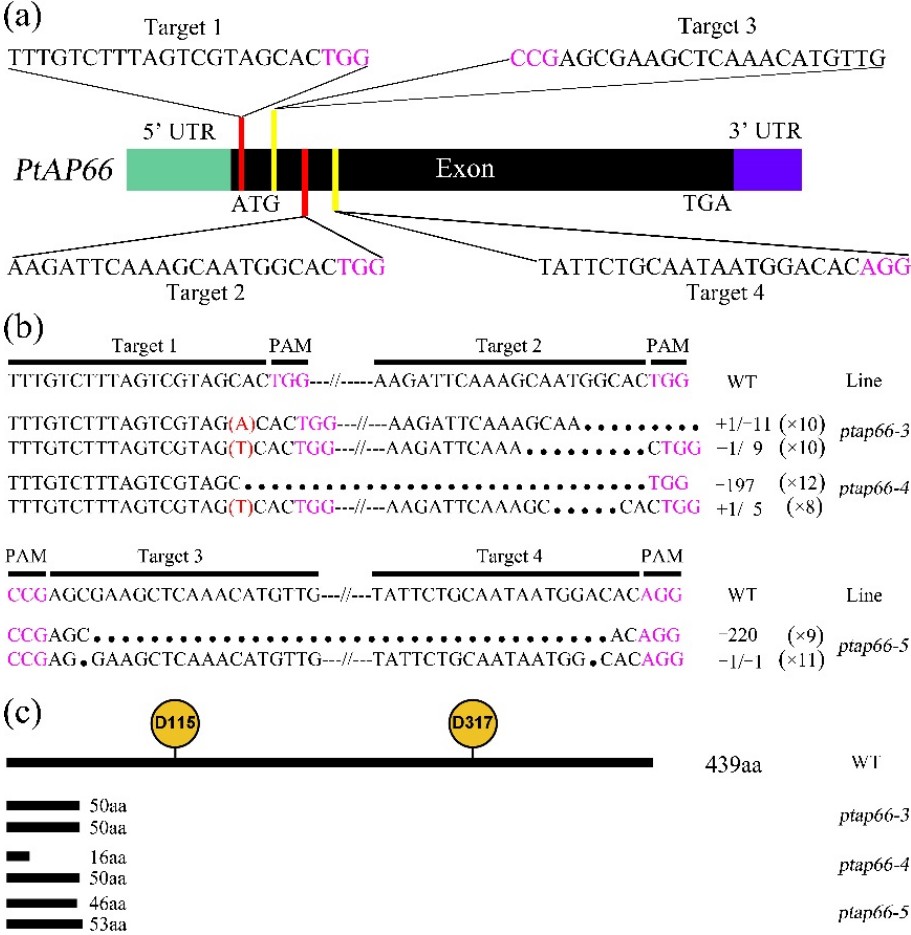

**Figure 3.** CRISPR/Cas9-induced *PtAP66* mutations in *P. trichocarpa*. (**a**) Diagram of two pairs of CRISPR/Cas9 target sites of *PtAP66*. The exon is displayed as a black box, while untranslated regions (UTRs) are displayed as coloured boxes. Target1/2 and Target3/4 indicate the positions of gRNA-targeted sites by different pHSE401-2gRNA vectors. (**b**) Mutations at the gRNA target site for *PtAP66* mutants. Black text denotes the native gRNA target sequences, purple text denotes the PAM (protospacer adjacent motif); dots indicate the deleted nucleotides; and red letters in parentheses represent the inserted nucleotides. The text on the right summarises the mutation details in three independent CRISPR/Cas9-induced lines. (**c**) Schematic representation of selective mutated PtAP66 proteins. The native protein is composed of 439 amino acids, and the highly conserved aspartic residues (D115/317) are indicated by the yellow circle. The amino acid numbers deduced by Cas9/gRNA induction are indicated to the right.

### 3.4. Loss of PtAP66 Decreases Wood SCW Deposition

To determine whether the mutations of *PtAP66* would affect the growth and development of *Populus*, we examined the phenotypes of mutants grown in the greenhouse. Compared with the WT plants, the *ptap66* mutants showed normal shoot growth over a growth period of 4 months (Figure 4a). Quantitative measurements showed no significant changes in the diameter of the stems, widths of various transverse stem tissues, number of internodes, lengths of internodes, or lengths and widths of leaves in the mutant lines compared with the WT plants (Figure 4b,c and Figure S2a–d).

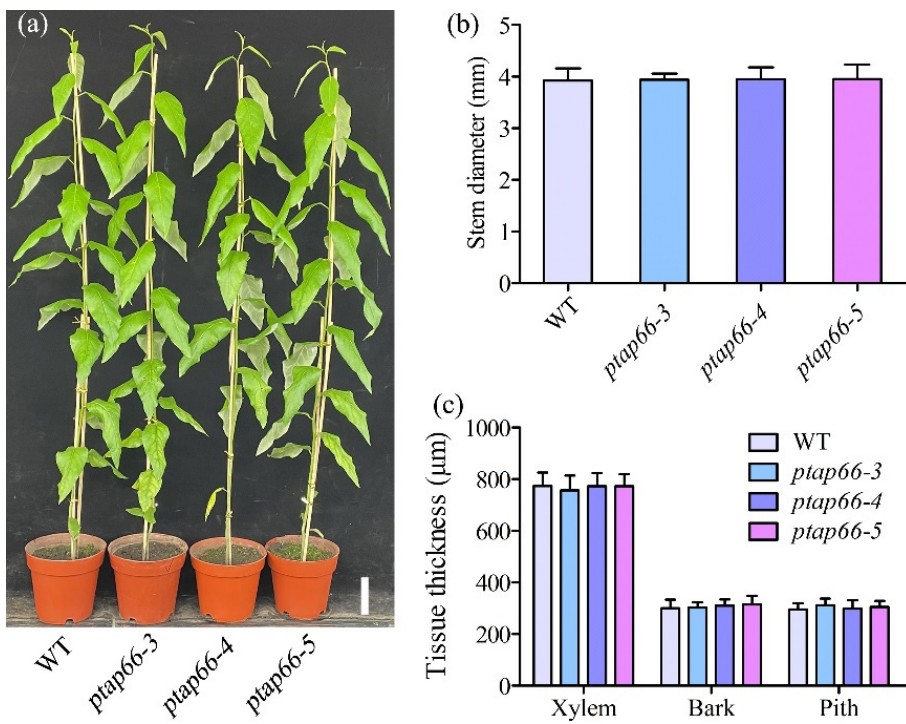

**Figure 4.** Characterisation of the *ptap66* mutant lines. Growth phenotypes of 4-month-old *ptap66* mutants and WT plants. Bar = 10 cm. These plants were measured for tree height (**a**), stem diameter (**b**), and stem tissue thickness (**c**). Error bars represent ± SD values of six biological replicates.

The SEM analysis showed that the loss of *PtAP66* decreased the SCW thickness of secondary xylem fibres (Figure 5a–d). The statistical analysis showed that the SCW thickness of mature fibres in *ptap66* plants decreased by 38.95–41.41% (Figure 5e) while the fibre size was not obviously changed compared with that of the WT plants (Figure 5f,g). Similar results were also observed in 3-month-old *ptap66* mutants and WT plants (data not shown). To further verify the role of *PtAP66* in SCW synthesis, the main wood components, including the cellulose and lignin contents, were examined in *ptap66* mutant lines. Compared with the WT plants, the cellulose contents in these mutant lines were decreased by 4.90–5.57% (Figure 6a), while the lignin contents did not show significant changes (Figure 6b). Overall, these results suggest that the knockout of *PtAP66* does not interfere with normal plant growth but affects SCW deposition in poplar wood.

### 3.5. Loss of PtAP66 Decreases the Expression of SCW Synthesis-Related Genes

We further examined the expression of SCW synthesis-related genes through an RT-qPCR analysis. Compared with WT plants, the SCW cellulose synthesis genes *PtrCesA4*, *PtrCesA7A*, and *PtrCesA7B* [39,50] were downregulated in the stem xylem of the *ptap66-3* line, which was consistent with the decrease in cellulose contents in wood (Figure 7). In addition, the expression of the hemicellulose synthesis-related genes *PtrGT47C* and *PtrCSLA1* [6,51] were also downregulated in *ptap66-3* plants (Figure 7). These findings reveal that the loss of *PtAP66* decreases the expression of SCW synthesis-related genes.

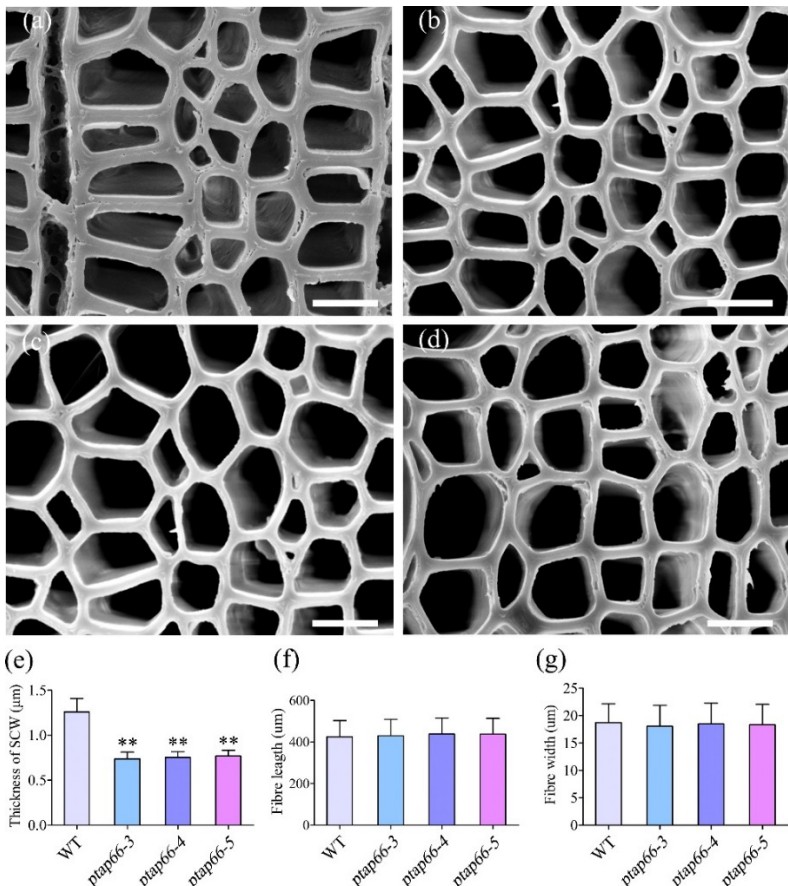

**Figure 5.** Effects of CRISPR-edited mutations in *PtAP66* on wood fibre anatomy. SEM images of representative mature wood fibre walls of IN12 in WT (**a**), *ptap66-3* (**b**), *ptap66-4* (**c**), and *ptap66-5* plants (**d**) at 4 months of age. Bar = 10 μm. (**e**) Statistical analysis of the SCW thickness of mature fibres in the xylem of *ptap66* mutants and WT plants. Statistical analysis of xylem fibre length (**f**) and fibre width (**g**) in the xylem of *ptap66* mutants and WT plants. Error bars represent ± SD values of three biological repeats. Asterisks indicate significant differences between each line of *ptap66* mutants and WT plants by Student's *t*-test (** $p < 0.01$).

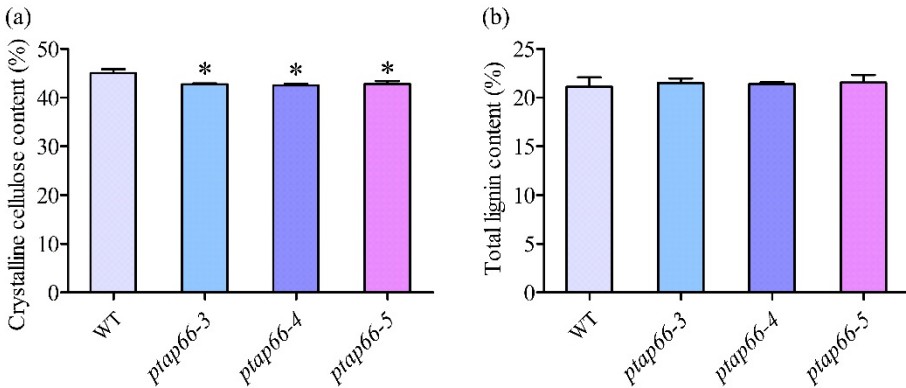

**Figure 6.** Analysis of crystalline cellulose and total lignin contents in dry wood of WT plants and *ptap66* mutants. Starch-free wood was used to measure crystalline cellulose (**a**) and total lignin (**b**) contents. Error bars represent ± SD values of three biological repeats. Asterisks indicate significant differences between each line of *ptap66* mutants and WT plants by Student's *t*-test (* $p < 0.05$).

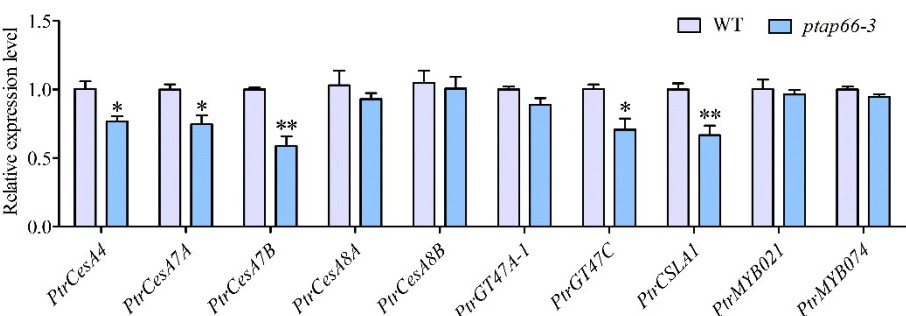

**Figure 7.** Expression analysis of genes associated with SCW synthesis in WT and *ptap66-3* plants. The *PtActin2* was used as an internal control. Error bars represent ± SD of three biological repeats. Asterisks indicate significant differences between *ptap66* and WT plants by Student's *t*-test (* $p < 0.05$ and ** $p < 0.01$).

## 4. Discussion

SCW deposition represents an important biomass accumulation process in terrestrial ecosystems. Although SCW deposition-related carbohydrate active enzymes and transcriptional regulation networks have been identified [5–7], few of the factors regulating SCW deposition have been identified. In this study, we characterised a PM-located atypical AP gene, *PtAP66,* which regulates wood SCW deposition in poplar. The defective phenotypes of the Cas9/gRNA-induced mutants showed that *PtAP66* played an important role in SCW deposition and that the expression of some SCW synthesis-related genes was compromised. Thus, atypical *PtAP66* is proposed to be involved in SCW deposition in trees.

Previous phylogenetic analyses revealed that PtAP66 and PtAP5 are grouped with the *Arabidopsis* ASPR1 (ATYPICAL ASPARTIC PROTEASE IN ROOTS 1) gene [30], which is mainly expressed in primary roots and mature pollen grains [52]. In *Populus*, *PtAP66* and *PtAP5* are highly and preferentially expressed in developing xylem, although *PtAP5* is expressed in xylem vessels and *PtAP66* is expressed in fibres [30]. In addition, ASPR1 localises in the endoplasmic reticulum in *Arabidopsis* [52] while PtAP66 localises on the PM (Figure 2). *ASPR1* knockout suppressed lateral root development [52], while the *ptap66* mutants showed reduced SCW deposition (Figures 5 and 6). Hence, the *PtAP66* gene does not represent a functional orthologue of the *ASPR1* gene in *P. trichocarpa*. Compared with the localisation of ASPR1, PtAP66 localisation might be related to amino acid sequence specificity. A plausible reason for this association is that the N-terminus of PtAP66 might include a putative transit peptide that mediates protein targeting, which is similar to other atypical APs [52,53]. Another is the glycosylation modification in PtAP66 [30], which might be vital for determining the protein targeting route [47].

To date, only one type of protease, cysteine protease (AtCEP1), has been implicated in SCW deposition, which was based on genetic evidence [18]. CEP1 is located in the xylem cell wall in *Arabidopsis*, and *cep1* mutant plants exhibit a thicker SCW and reduced xylem cell numbers; moreover, a large number of SCW-related genes encoding enzymes and transcription factors for cellulose, hemicellulose, and lignin synthesis are upregulated in *cep1* mutants [18]. In this study, loss of *PtAP66* did not lead to significant changes in the xylem width and fibre size (Figures 4c and 5f,g), implicating its feeble role in xylem cell proliferation and expansion, which was consistent with the expression pattern across the cambium and xylem expansion zone of poplar stems (Figure 1). However, the decreased fibre SCW thickness and SCW synthesis-related gene expression in *ptap66* plants imply that *PtAP66* plays an important role in wood SCW deposition (Figures 5 and 7). This cellular and molecular evidence indicates the differential role of *PtAP66* to *CEP1*, suggesting that *PtAP66* and *CEP1* might be involved in different pathways that regulate SCW deposition. Additionally, the promoter region of *PtAP66* has two SNBE (secondary wall NAC binding element) *cis*-elements [30], suggesting that it might be an intermediate regulator acting on SCW-associated genes [54]. Further studies are needed to reveal its upstream regulators.

SCW deposition is mediated by signal transduction, as suggested by defective mutants. For example, *AtVRLK1* (vascular-related receptor-like kinase 1) functions as a signalling component for coordination of SCW thickening in xylem cells and overexpression of *AtVRLK1* results in a thin SCW [17]. Inhibition of NADPH oxidase could decrease secondary wall deposition by reducing the production of ROS [55]. Loss of *PtAP66* resulted in SCWs with thinning fibres and decreased the expression of several SCW synthetic genes in secondary xylem (Figures 5 and 7), suggesting that *PtAP66* might mediate signal transduction, thereby promoting SCW deposition. We propose that PtAP66 is involved in the proteolysis pathway or activation of the PM signalling pathway in xylem fibres to regulate SCW deposition.

Knockouts of several atypical *APs* cause distinct growth and development defects. For example, loss of function of *ASPR1* decreased primary root growth and lateral root number in Arabidopsis [52]; The Arabidopsis *aspg1* mutants displayed enhanced seed dormancy and reduced seed viability [29], and disruption of *A36* and *A39* in Arabidopsis resulted in higher abortion ratios in siliques [56]. However, in *ptap66* plants, we did not detect any obvious differences in growth and development compared with the WT plants. In contrast, reductions in the thickness of SCW fibres and wood cellulose contents were detected in *ptap66* plants (Figures 5 and 6). Thus, the function of *PtAP66* is different from that of other known atypical APs, which is possibly due to their differential specific substrates. To date, the substrate of atypical APs has not been reported. Therefore, future efforts to identify the substrate of PtAP66 in secondary xylem will provide insights on the molecular mechanism underlying the involvement of PtAP66 in wood formation.

## 5. Conclusions

In this work, atypical aspartic protease *PtAP66* was highly expressed in xylem fibre that was suggested to be a PM-localised protein. We used CRISPR/Cas9 technology to produce *ptap66* mutant trees. Genetic, biochemical, and transcriptional analyses indicated that *PtAP66* is involved in wood SCW deposition. The decrease in SCW deposition in *ptap66* mutants implies that *PtAP66* is a potential gene for molecular breeding to improve tree biomass production. Additionally, whether *PtAP66* affects the programmed cell death of fibres during xylem maturation remains an interesting question for investigation. Hence, future studies should focus on identifying the actual substrate of PtAP66 to elucidate its specific role in SCW deposition during wood formation.

**Supplementary Materials:** The following are available online at https://www.mdpi.com/article/10.3390/f12081002/s1, Figure S1: Multiple sequence alignment of PtAP66 with some APs in plants; Figure S2: Phenotype analyses of 4-month-old WT and the *ptap66* mutant plants; Table S1: All primers used in this study; Table S2: The Cas9/gRNA-targeted mutations in *PtAP66* gene.

**Author Contributions:** Conceptualisation, S.C., Y.C. and C.Y.; methodology, S.C.; validation, S.C., H.J. and C.W.; investigation, S.C., H.J., C.W., M.G. and J.C.; writing—original draft preparation, S.C.; writing—review and editing, Y.C. and C.Y.; visualisation, S.C. and H.J.; supervision, Y.C. and C.Y.; funding acquisition, Y.C. and C.Y. All authors have read and agreed to the published version of the manuscript.

**Funding:** This work was financially supported by the Fundamental Research Funds for the Central Universities (2572018BW01), the Heilongjiang Touyan Innovation Team Program (Tree Genetics and Breeding Innovation Team), the 111 Project (B16010), and the National Natural Science Foundation of China (31770637).

**Acknowledgments:** We thank Na Guo (NEFU) for her assistance in figure format and discussion.

**Conflicts of Interest:** The authors declare no conflict of interest.

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
