# Peer review of "Functional Characterisation of the Poplar Atypical Aspartic Protease Gene PtAP66 in Wood Secondary Cell Wall Deposition"

_forests, doi:10.3390/f12081002_

Round 1

Reviewer 1 Report

(My exact decision is ‘More than “Minor” but less than “Major” Revision’.)

  This manuscript focuses on the poplar atypical aspartic protease gene PtAP66. Examinations of when it is expressed in terms of secondary cell wall (SCW) formation, of its subcellular localization, and of how mutations of PtAP66 would effect anatomically and chemically on phenotype etc. were conducted. They found that PtAP66 transcripts increased dramatically in xylem expansion part and maintained high levels in the SCW formation zone; PtAP66 is suggested to be localized in the plasma membrane; and a decrease of SCW thickness and crystalline cellulose content in ptap66 mutants was observed. The data is presented clearly and analyses were properly conducted.

  However, I felt that the logical structure and tone of the manuscript seemed to be that the (hypothetical) conclusion which PtAP66 is a novel regulator of SCW formation comes first, and that results were interpreted basically by fitting the context for this conclusion. I think (at least in the manuscript structure) it should be vice versa. The result that the difference of mutants in phenotype observed were not large amplified the impression. There might be others which have greater impact on SCW formation. If possible, please revise the manuscript from this point of view, to make your manuscript more acceptable for the readers.

1) I know the detailed description of some experimental methods are in the references. But please include some of key words and/or factors and short description for the convenience of the readers. Roughly what was the amount of sample used for compositional assay, how did you measure fiber length?

2) I am not used to the styles of section titles in the “Results”. If it is not the custom of this Journal, I would like to advise to revise.

3)How did you come up choosing 4-month-old for the analysis? Do you have any perspective results if you continued to grow them? (I am asking this because, as I have mentioned already at the top, the difference observed between the mutants and the wild type was little.)

4) Do you have any (hypothetical) idea what has increased instead of crystalline cellulose?

Also, there are several minor comments:

5) (p.1 l.36) Regarding the abbreviation “PCD”, it appears one time only at the very end of the manuscript. I recommend to stop using the abbreviation and spell the words out in the “Conclusions”.

6) (p.1 l.43) “l” seems to be not necessary. (Isn’t it should be “enzymes”?)

7) (p.2 l.72) It is a little difficult to catch what you are referring by the words “this group”. From the context, I thought it was “woody plants”, but literally, it might seem to be “herbaceous plant”.

8) (P.4 l.89) Are you using the letter “delta”? It seems to me that they are triangle marks.

9) (P.6 l.253) I assume typography (?) of the words in the parentheses should be uniform.

10) (P.7 Figure 3) The blue color used for the “blue letters” are too dark to distinguish (recognize). Please revise and improve visibility.

11) (P.9 Figure 5) Regarding the Y-axes in (f) and (g), the word “fiber” is spelled in American English style. Please standardize to British English.

12) (This comment is common to Figs. 5 and 6) I think there is no need to write a description for symbols that are not used in each figure. (Specifically, * for Fig.5 and ** for Fig.6.)

Reviewer 2 Report

Comments can be found in the enclosed file

Round 2

Reviewer 1 Report

Thank you for your sincere consideration and response to the previous comments.

Not about the content itself, but I still have a technical concern. Regarding Fig. 3, the choice of the colors they have chosen it still very difficult to differentiate (low visibility). Please consider advising from the Editorial a color combination that will improve visibility. I am afraid that the present choice is much difficult for colorblind people to distinguish.

Reviewer 2 Report

The manuscript has been revised according to suggestions provided with an improvement in the final quality.
